# An "Interactive Learning Model" to Enhance EFL Students' Lexical Knowledge and Reading Comprehension

Lei Yang

School of Foreign Languages, Xi'an Jiao Tong University, Xi'an 710049, China; selina.lei@mail.xjtu.edu.cn

**Abstract:** (1) Background: The ability to read successfully in the context of college English as a foreign language contributes to sustainable language acquisition and academic development. (2) Research problems: To enhance the sustainability of reading, the article proposed the new teaching model-interactive learning model. What is the impact of the "interactive learning approach" on EFL learners' content and vocabulary learning? (3) Methods: "Learning Model" via the experiment class and the control class in two posttests: immediate posttest and three-week posttest. In the experiment class, students were taught with the "Interactive Learning Model" and students in the control class were instructed with a traditional approach without student interaction. (4) Results: The results of the statistical analyses indicate that the interactive learning class and the control class performed similarly on both the content and vocabulary tests in immediately posttest; but in the delayed posttests, the interactive learning class obviously outperformed the traditional class, that is, the students in the experiment class forget less vocabulary and content learning through intra/inter group discussion. (5) Significance: The significance of the research demonstrates the "Interactive Learning Model" improves students' language learning motivation and offers the benefit of processing the foreign language more deeply and internalizing their knowledge through implicit learning.

**Keywords:** college English learning; teaching methodology; interactive learning model

## 1. Introduction

Developing a sustainable society relies heavily on broad measurable elements (e.g., economic factors), however, human choice and human needs are at its core [1,2]. A comprehensive examination of human-related aspects of sustainability reveals that a human learning environment and effective communication are key drivers of sustainability [2], relying on language and literacy development. Reading is one of the important language skills, empowering individuals and enhancing societal development. Modern society relies heavily on reading to promote individual development [3]. As the information age advances, reading development is not restricted to the mother tongue, and efforts are being made to facilitate reading in foreign languages as well.

However, historically, English as a foreign language (EFL) reading instruction has not been particularly effective in accomplishing a series of challenging learning tasks because EFL students have been taught to match the subject-verb agreement. There is usually little interaction and discussion among students about what is being read. Thus, students often consider it boring to read in English and find it hard to comprehend the text [4]. If there is lack of interactive learning among students and students are rarely asked questions, and they are not given the chance to think and participate, the classroom is very boring, and students are not interested [5]. Therefore, it has been proposed that verbal interaction is the basic condition of second language acquisition [6].

In the "Interactive Learning Model", we refer to the interaction theory of language learning [7,8] as well as the socio-cultural theory of learning [9,10]. Students interact both intra-group and inter-group after reading, both orally and in writing, ensuring a social atmosphere. Through exposure to the language environment and use of EFL in oral

interactions, they develop the ability to communicate and send meaning in oral interaction. Essentially, the "Interactive Learning Model" is an attempt to build "shared responsibility" or "shared intentionality" by creating a "knowledge gap" of content and language [11–14]. Interaction is one of the characteristics of socially shared knowledge [15]. Content and language learning are increased by bridging the knowledge gap [16].

## 2. Review of the Literature

The "Interactive Learning Model" integrates three approaches or theories in reading: reading interaction theory, sociocultural interaction theory, and lexical learning theory. Each one will be reviewed in the literature section.

### 2.1. Interaction Hypothesis Theory and Comphreshesion Input

In terms of second language acquisition, Long's (1981, 1996) Interaction Hypothesis is one of the most influential systematic theoretical formulations. It has contributed to what is now called the interactionist approach [7,17]. The interactionist approach focuses on conversational interaction in language learning because conversational interaction involving three elements: input, output, and meaning negotiation makes language learning more likely to be happen.

According to Krashen (1981), input (or comprehensible input) is helpful to acquire a second language. However, input alone can be insufficient; the developing language knowledge requires output and input [18]. Krashen (2017) emphasizes comprehension reading (input) is an effective method of improving reading, since comprehension may facilitate language acquisition [19]. Additionally, Swain emphasizes the output beyond the user's current linguistic level by using new forms in the second language so that the learners are able to noticing gaps in their L2 knowledge [16]. Therefore, efforts are then made to create better conversational expressions in the second language.

The negotiation of meaning refers to the attempt by speakers in conversation to clarify information. It is an important process for language learning as it enables learners to acquire language knowledge they lack through modifying their input/output [20]. As part of meaning negotiation, learners receive feedback in a variety of formats and have the opportunity to adjust their output accordingly. For example, the performance on standardized English tests was more closely tied to comprehension reading with negotiation than skill building (referring to conscious learning, output practice, and correction) [21]. Therefore, the Interaction Hypothesis had a major influence on foreign language teaching and learning in the classroom context. Despite this, researchers realize that there are still a number of problems with the Interaction Hypothesis [22], for example, cognitive development can occur only inside the head of an individual, separating linguistic competence and learning from social interaction; the context was simplified into independent variables affecting the cognitive process. Interactional approaches have been urged to shift focus from internal cognitive and psychological aspects of interaction to external social mechanisms and processes [20]. According to sociocultural theory of human learning to be discussed in the following part, interaction is a social process heavily influenced by sociocultural factors [9].

### 2.2. Sociocultural Interaction Theory

From an early age, humans learn to interact with others through language, as Vygotsky (1978) noted. In this sense, social interaction is crucial to cognitive development [9]. Thus, social interaction plays a crucial role in cognitive development. In Vygotsky's theory, "zones of proximal development (ZDF)" are also important constructs. A ZDF is defined as "the distance between an individual's actual developmental level as determined through independent problem solving and his or her potential development", that is, the distance between the actual developmental level and the level of potential development [9]. Vygotsky's theories stress the fundamental role of social interaction in the development of cognition, as he believed strongly that community plays a central role in the process of "making meaning" [9]. When scaffolding support is provided from a more knowledgeable and capable

individual, a learner can achieve potential learning growth. In order to maximize learners' potential, a teacher needs to provide them with learning experiences. By creating meaningful communication opportunities in the target language, teachers give learners the chance to engage in meaningful communication. That is to say learners must engage in meaningful communication to fill in gaps in their content and language knowledge through the use of learning activities or tasks, for example, using oral communication to discover what they don't know. It is vital to take responsibility for filling these knowledge gaps collectively and individually, as it facilitates smooth communication and result-driven learning [10]. Furthermore, for effective learning, it is necessary to construct a similar social setting including activities not only within the group, but also between groups [7]. Within-group and between-group activities are sometimes needed because the learners in a group may not have the content and language knowledge to communicate effectively. The students participated in a group discussion in a given group after reading the material because there was a lack of relevant content and linguistic knowledge. Furthermore, they will exchange information with other groupmates to gain additional information. Therefore, in order to successfully complete their communicative activities, both between and within-group activities are sometimes needed [23]. In short, in sociocultural theories of learning, social interaction is highlighted as the key dimension of the interaction approach, emphasizing the significance of sociocultural factors through interactional activities. During interactions, the need to bridge knowledge gaps drives the development of shared responsibility.

*2.3. Interactive Reading Instruction Approach*

In order to help foreign/second language learners improve their reading ability, three of the most relevant interactive teaching approaches are promoted: "jigsaw reading"; "read-ask-tell"; and "E-plus approach" [11,24,25].

The "jigsaw reading" approach emphasizes that each part of the text to be read in a class is assigned to a group of students, each of whom must read and discuss it together to become experts on its information. A new group requires students to first teach their part of the text, then pass it on to the next.

In the "read-ask-tell" approach [25], a similar procedure is used, but it is designed mainly for the reading of newspaper articles and assigning students to groups according to the level and length of the newspaper articles. In the asking stage, first, the students do "unstructured informal asking" where they ask for help regarding any vocabulary, concept, and content difficulties they may have. Then they take part in "structured formal asking" where they discuss answers to questions assigned by the teacher, "trying to reach a consensus of opinions" [25]. Finally, each group elects its spokesperson. In tell stage, the spokesperson in each group will practice explaining the group's article in summary form, then go to the other groups rotationally to explain the article and answer any questions. The two approaches have something in common: students need to fill in information gaps through meaningful communication and problem-solving.

According to Boutorwick et al. (2019), the "ER-plus approach" has the following steps [11]. First, students read graded reader books. When the students have completed the reading, they are assigned to a small group and perform a 15-min "Say-it" activity. Each group member chooses a prompt from a three-by-three grid to discuss the graded book they have just read [11]. Through this activity, students can practice and reinforce what they learned from their individual reading.

Despite the differences in procedures and activities, all three approaches emphasize interactive discussions among students to facilitate reading comprehension. Interaction learning theory and sociocultural theory of learning are both driven by or reflect the importance of meaningful student interaction.

*2.4. Lexical Learning Theory and Reading*

Reading has been proven to increase lexical learning, particularly incidental lexical learning, in numerous studies [11–13,26,27]. In various research designs and learning

achievement measures, reading can have a positive impact on lexical learning. It has been shown to improve both the quantity and quality of lexical learning, as well as breadth and depth. Moreover, researchers have found that learners are better able to learn a word when they encounter it more frequently in reading [11].

The "Interactive Learning Model" is based on theoretical and practical results from academia. This study focuses on the language environment of China's foreign language teaching and the college students who have received traditional classroom instruction for many years. As is said in the beginning, the "Interactive Learning Model" focuses on creating a "social" or "communicating" setting, in which "meaningful" interaction is conducted by oral communications with others. The "shared responsibility" and "shared intentionality" are caused by the "knowledge gap", both on the contents and the language. Through "meaningful" interaction, they facilitate oral communication [28]. The essential part of the "Interactive Learning Model" can be shown in the Figure 1. Implementation details may vary depending on specific circumstances and class requirements.

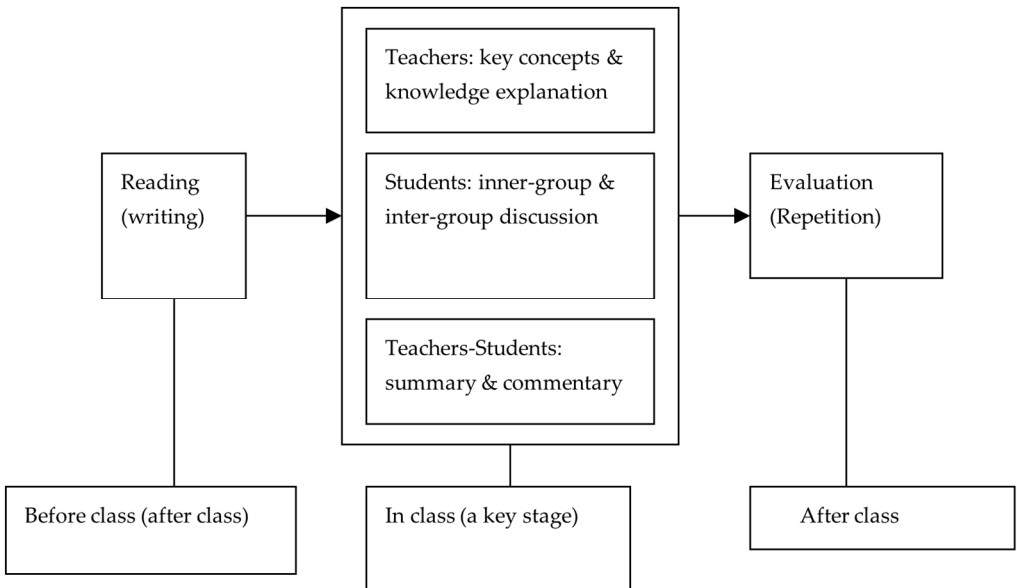

**Figure 1.** The "Interactive Learning Model".

As is shown in Figure 1, the "Interactive Learning Model" first and foremost emphasizes the succession of the Interaction Hypothesis. The central role is played by oral discussion and communication. In addition, to some degree, it made a breakthrough to the idea of the Interaction Hypothesis, which separated linguistic competence and learning from social interaction [24]. It is manifested in two aspects of the model. Initially, reading is a part of learning, even as the starting point, because it prepares the student for summary writing and inner-group/inter-group discussion Meanwhile, the reading should have certain breadth and depth and be based on contents. The content-based reading also represents another aspect of the "Interactive Learning Model", that is, apart from emphasizing social interaction, it also emphasizes "meaningful" communication between/within groups to support reading comprehension in-depth (surely the deep level processing of the foreign language input is involved). To achieve a deep understanding and extension of reading content in the classroom, it is necessary to emphasize the construction of a similar "society context", composed of activities within the group and activities between groups [8]. The "knowledge gap" of content and language construct "common responsibility" or "common intention", that is to say, the individual alone should not be responsible for their "knowledge" within their rights or obligation, but also the group members to ensure that interactions go smoothly [29].

"Common responsibility" or "common intentionality" driven by "the knowledge gap" may contribute to two important "episodes" in the development of foreign language

proficiency: LREs (language-related episodes) [16]; and IR (information retrieval) [16,23,30]. Swain and Lapkin (1995, 1998) define LREs as "any part of a dialogue where the students talk about the language they are producing, question their language use, or correct themselves or others" [16,31]. In most cases, when LREs occur, students work together to solve the knowledge gap without a teacher, which creates many opportunities for language use, such as correcting, suggestions, self-talking, etc. IR, information retrieval, is the other important episodes. Jakonen and Morton (2015) found that when students were committed to a reading task about a certain topic (such as a historical topic), without information to continue the task—either linguistically, such as word meaning, or content, such as some specific facts [23]—the group members asked the peers and neighbors for information, which ignited the "Epistemic Search Sequence" (ESS). Finally, they worked together to solve the problem.

According to the introduction above, students are at the center of the "Interactive Learning Model", and teachers are the source of "scaffolded help" [8,32], including providers of high-quality language input, explainers of core concepts and knowledge, key coordinator in class, and interpreter of key literature. The teaching in the "Interactive Learning Model" is a kind of content and language integrated learning (CIL), which focuses on training students' reading, writing and oral communication skills. Learning resources and language resources can be triggered via LREs and IR to provide opportunities for learners to develop their language skills. It also takes into account the unique characteristics of college English teaching in Chinese and foreign language environments. The objects of college English teaching in China are adults, and they have spent 8–10 years on foreign language grammar learning and training in the classroom environment. The "Interactive Learning Model" that integrates content and language has tried to discard repetitive boring teaching, and either increase the practice of knowledge of the English language (grammar knowledge) or attempt to continue to stimulate the motivation and interest in foreign language learning, and provide lasting motivation for language development [33].

To study the impact of the English teaching mode on the language skills and emotional factors of Chinese college English learners, this article will use a comparative experiment to analyze its impact on the lexical development of Chinese college English learners. Lexical knowledge is closely related to the use of second language, so how to effectively expand the lexical knowledge of second language learners has always been the researcher's focus. Hereby, we posed the following research questions.

1. What is the impact of the "interactive reading approach" on EFL learners' content learning, as measured by immediate and delayed posttests?
2. What is the impact of the "interactive reading approach" on EFL learners' vocabulary learning, as measured by immediate and delayed posttests?

## 3. Methodology

To examine the effectiveness of the "Interactive Learning Model" via the experiment class and the control class, the study invited 40 non-English major graduate students in the experiment and control class. In this class experiment, one class was taught with the "Interactive Learning Model" and the control class was instructed with a traditional approach involving no student interaction. After the experiment, students in two classes took two posttests: immediate posttest and three weeks later posttest and were interviewed after immediate posttest.

### 3.1. Experimental Design

A two-group posttest design was used in this study (an experimental group and a control group). Experimental groups were taught using an interactive reading approach that included reading, group discussions, and writing after reading. The interactive reading approach drew on practices from the various interactive reading approaches, such as "jigsaw reading", "read-ask-tell", and "ER-plus". Besides group discussion and writing activities, the interactive reading approaches were driven by filling information/knowledge

gaps similar to those found in the "jigsaw reading" and "read-ask-tell" approaches. A traditional teaching approach was used for the control group based on the practices and procedures typically used in a traditional classroom, including student individual reading, teacher lecturing, and questions and answers.

*3.2. Participants*

Students from two parallel classes in Xi'an Jiao Tong University participated in this teaching experiment, which was classified into experimental class and control class. The students in both classes (experiment class and control class) took the placement test before they enrolled in English courses They were A level in the placement test (90 scores and above in the placement were in A level; 80–90 scores were in the B level; and the remaining students were C level). Each class had 40 students, who were non-English major graduate students. The male to female ratio in each class was approximately 3:1, and the average age was approximately 22 years (range: 21–24) (see Table 1). As Chinese students, they spoke Chinese as their native language. They all started learning English in the third grade (7–8 years old) of elementary school. Years of English learning was about 13 years, and they had all passed the College English Test Level 4. At the time of the experiment, they were taking English writing classes taught by the same teacher. The teacher, with a doctorate degree, had extensive experience teaching writing. Judging by the comprehensive English test scores of the two classes in the previous semester, the English level of the two classes was very close ($t = -0.32$, $p = 0.75$).

**Table 1.** Basic information of participants in the experiment group and the control group.

| Groups | Age | Years | Test Scores (100) |
| --- | --- | --- | --- |
| Experiment group | About 22 years old (21–24) | 13 | 85.7 (3.95) |
| Control group | About 22 years old (21–24) | 13 | 85.9 (3.02) |

*3.3. Materials*

To effectively implement the "reading and discussing" English teaching model, we first carefully considered the content of reading in the teaching experiment, then, we decided on Sections 2 and 3 of Florida Travel Manual (2018 edition) (handbook for short) [34] as classroom reading material for the teaching experiments for four reasons: (1) Based on Vygotsky's Zone of Proximal Development and Krashen's i + 1, the researchers ensured that the difficulty of reading materials allowed students to "hop it up" and it met the experiment requirement. (2) Students will gain some background information from this manual, as well as enhance their interest. (3) The manual, compiled by the Florida Department of Tourist Management, USA provided various color graphics and signs related to transportation and language descriptions with real and vivid scenes. (4) The manual is an authentic English text. The vocabulary level is appropriate, because 85.6% words in the manual are from the General Service Word List (GSL), including the 2000 most commonly used English words. Students took the College English Test (Band 4) administered by the Ministry of Education, which is a required English test for college students. In accordance with the Ministry of Education's information about CET-4, passing this exam requires a vocabulary size of 4200. It means that the manual is appropriate for the participants and the participants have a high enough vocabulary.

The manual has ten parts, 80 pages. The researchers chose two parts about traffic rules as experiment material: the second (Section 2—Signals, Signs and Pavement Markings); and the third (Section 3—Safe Driving), 3800 words, 18 pages, considering the limitation of class time. In the two parts, researchers picked up 10 words as target words, generally used in daily driving.

Researchers first used online word frequency statistics software Write Words to perform word frequency statistics and selected the ten most frequently occurring words (excluding a, the, and other function words, considering that they can appear in high

frequency in various parts of the reading material) (see Table 2). Each of these words occurred between 19 and 48 times in 18 pages. The ten words are from GSL. Second, the researchers replaced these ten words with ten pseudo-words, and at the same time strictly ensured that no more than two new words would appear around the sentence after the pseudo-word's replacement, so as to avoid comprehension difficulties for the participants. For example:

1.  Traffic signals apply to drivers, motorcycle riders, bicyclists, moped-riders and **spoists**.
2.  Red light: At a red light, come to a complete stop before you reach the **courch**, stop line or crosswalk. Remain stopped unless turns are allowed on red.

**Table 2.** Target words, pseudo-words and frequency of words.

| Words | Contents |
|---|---|
| Target words | Intersection (32), Ramps (21), Pedestrian (40), Roundabout (29), Lane (33), Curve (19), Hubs (31), Brake (48), Flash (37), accident (32). |
| Pseudo-words | Intersection = Courch; Ramps = Rroff; Pedestrian = Spoist; Roundabout = smunth; Lane = pring; Curve = kwuk; Hubs = Smebbs; Brake = Brousted; Flash = Fleem; Accident = Acoord |

The "spoist" in example 1 and the "courch" in example 2 are target pseudo-words. Generally speaking, these pseudo-words in sentences do not fundamentally change the structure of the sentence and also keep the original style. It is a common paradigm for researchers to use pseudo-words instead of real words to carry out research [11,29]. The pseudo words we used followed other research, and were designed to look and sound totally different from their real-word counterparts to avoid the students guessing their meanings [35,36]. In the experimental class, the 18 pages of material were divided into six parts from the beginning to the end. A six-part reading text (each with three pages plus the cover page of the manual) was then distributed evenly among the six groups. Therefore, the 40 students in the experimental class were divided into six groups (seven in each of the first four groups and six in each of the fifth and sixth groups). The 18 pages of material were divided into six parts from the beginning to the end. In other words, the material of each group is different from the other groups, which naturally causes a "knowledge gap" among the groups. Each pseudo-word appears three times in each set of material on average, total 18 times in six sets of materials. In the control class, students were not divided into any groups. They all received the entire 18-page reading text, unlike students in experiment class, who were given parts of the reading text. The 18-page reading text in the control class was the same as in the experiment class, with pseudo-words replacement.

*3.4. Experimental Process*

3.4.1. The Experimental Process in the Experimental Class

The process in the experimental class (Table 3) is divided into four steps: (1) Reading. Participants took about 15 min to finish reading the materials from beginning to end without looking up words in a dictionary (2) Inner-group discussion and interaction. After finishing reading, members in each group discussed what they had read. The discussion was comprehensive, including both language issues, such as words, phrases, sentences, and grammar, as well as content, such as the traffic rules involved in the reading materials. (3) Summary of the reading materials. Participants could take notes during the discussion, and each group formed a summary of the content after the discussion. The discussion should have been completed in 20 min. (4) Inter-group discussion and interaction. After the group discussion, two members from each group were selected to rotate and communicate with members of other groups. This cycle continued until each member in each group had communicated with the other group members, and each group had completely understood the 18-page Manual. When communicating between groups, members in each group emphasized communicating in English, carrying their notes instead of printed reading

material. The teacher monitored the whole process, ready to answer student questions. The experiment took about 65 min.

**Table 3.** The difference between the experiment class and the control class.

| Teaching Classes | Teaching Models/Time | Materials | Numbers | Experiment Procedure (4 Steps) |
|---|---|---|---|---|
| The experimental class (A level) | Interactive Learning model (65 min) | FTM (2018) | 40 students) (6 groups) | Reading → Intra-group discussion → Summary → Inter-group discussion |
| The control class (A level) | Traditional teaching model (65 min) | FTM (2018) | 40 students (no groups) | Reading → Asking questions → Explaining → Writing |

Notes: FTM (2018): Florida Travel Manual (2018 edition) [34].

### 3.4.2. The Experimental Process in the Control Class

The experimental process in the control class (Table 3) is also summarized in four steps: (1) Reading. Students read the materials without looking up words in dictionaries. As time was unlimited, students were allowed to read repeatedly. The average time for each student was about 30 min. (2) Asking questions. While reading, students asked the teacher for assistance about both content and language of the reading material, such as words, phrases, sentences, or grammar. (3) Explanation. In accordance with traditional teaching methods, the teacher explained what they had read. (4) Writing. The students were required to write a summary in 20 min, describing the material they had read, and hand it in. The whole process also took about 65 min.

Before the experiment was conducted, the teacher clearly told all students in the experimental class and the control class: (1) Read the materials carefully, paying attention to the content and language (2) They would take two tests after reading, one being a vocabulary test, the other being a traffic regulation test (content test).

### 3.4.3. Experiment Material

The experiment material included the content test and the vocabulary test. Signals, signs and pavement markings, and safe driving were the topics of the content test. This test contained 20 items: ten true test questions and ten filler questions. Of the ten true questions, five were True or False questions and five were multiple choice questions.

Two sections were included in the vocabulary test, which aimed to test understanding of the meaning of words. For the first section, students had to translate each vocabulary item into Chinese. This section contained 20 items, ten of which were target pseudo-words and ten of which were real English words selected randomly from the text [35,36]. In the second section, students had to fill in the blanks in a sentence using a list of 40 appropriate words, with ten requiring target pseudo-words and ten requiring real English words. The remaining words served as fillers but were not included in the scores analysis.

### 3.5. Interview

For the retrospective interview, we randomly selected ten students from each class after the immediate posttest. Students were asked how they felt about the instruction and learning activities they experienced in class, and what effect what effect they believed the learning activities had on their learning.

In summary, the experimental class works as the "interactive learning" teaching model, and the control class served as the traditional "reading and teacher instruction" teaching model.

### 3.6. Data Collection

Two tests were performed in the experiment: one was performed immediately after the experiment was completed, and the other was performed after three weeks. Two test items were used in the experiment: the vocabulary test and the contents of the manual

(relevant traffic regulations) test. In the vocabulary test, students were first required to translate English words into Chinese. There were a total of 20 test words, ten of which were designed as pseudo-words and the other ten were true word as noise words, not included in the total score. These true words as distractors (fillers) were not included in the total scoring. Secondly, the students were asked to write an explanation or definition of the English word for a total of 20 test items, of which ten were pseudo-words and the other ten were true words as noise items, excluded from the total score. The content in the second test is exactly the same as the first one. The difference between the two tests is that the order of the first and second tests is reversed, and the order within each question is also disrupted to avoid repetition and familiarity to the greatest extent. A correct Chinese meaning in the first question or an accurate explanation or definition in the second question is given 1 point. Partial correct answers are given 0.5 points, and the wrong answers are 0 points. The total score is 20 points.

In the contents of the manual test, there were 11 questions: six True or False questions and five multiple-choice questions. There were also 11 noise questions, of which five were True or False questions and six were multiple-choice questions. The content of the second test was the same as that of the first test. The difference between the two tests was that the order was completely disrupted to eliminate the repetition effect. Regardless of True and False questions or multiple-choice questions, the right answers were given 1 point, and the wrong answers were given 0 points. The total score was 11 points. The reliability of this vocabulary test and the content test were 0.89 and 0.83, respectively, (Cronbach's alpha test).

## 4. Results

### 4.1. Results of Target Words Test

The results were analyzed using R language. Table 4 shows the descriptive statistics of the target words test (vocabulary test) and the content test results in immediate posttest and three-week posttest of the experimental class and the control class.

**Table 4.** Descriptive statistical results of target words and content test (mean and standard deviation).

| Groups | Time | Target Words Test (20 Points) | The Content Test (10 Points) |
|---|---|---|---|
| Experiment class | Immediately | 10.67 (4.98) | 7.53 (2.03) |
| | After three weeks | 6.13 (3.98) | 5.80 (1.53) |
| Control class | Immediately | 10.37 (4.54) | 7.33 (2.51) |
| | After three weeks | 3.50 (1.72) | 3.43 (1.81) |

It can be seen from Table 4 that the mean score of the target words test in the experimental class and the control class were very close (10.67 vs. 10.37, $p = 0.81$, see Figure 2) in the test conducted immediately after the end of the teaching experiment. However, in the delayed test (three weeks), the mean score of the target test in the control class and experiment class were very different (6.13 vs. 3.50, $t = 6.10$ $p < 0.001$, see Figure 3). A big drop in the mean scores in the control class indicates the influence of time on the control class more than the experiment class.

To investigate whether there was a significant difference of vocabulary scores between the immediate test and the three-week test for the experimental class and the control class, and whether the time had affected them, we used the ezANOVA ( ) function in R language to conduct 2 × 2 mixed designs ANOVA. The group was an inter-group factor with two levels (experimental vs. control class), and the time was an intra-group factor with two levels too (immediate test vs. three weeks test). The results showed that the group had no main effect (F (1, 78) = 2.02, $p = 0.21$, $\eta^2 = 0.016$), without considering other factors such as time. There was also no difference in the target words test results between the experimental class and the control class. However, time had the main effect (F (1, 78) = 576.50, $p < 0.001$); the immediate test scores were significantly better than the delayed test scores. Furthermore,

there was a significant interaction effect between group and time (F (1, 78) = 4.69, $p = 0.03$, $\eta^2 = 0.021$), and the effect of time on the control class was significantly stronger than the experimental class. These results can also be shown in Figure 4.

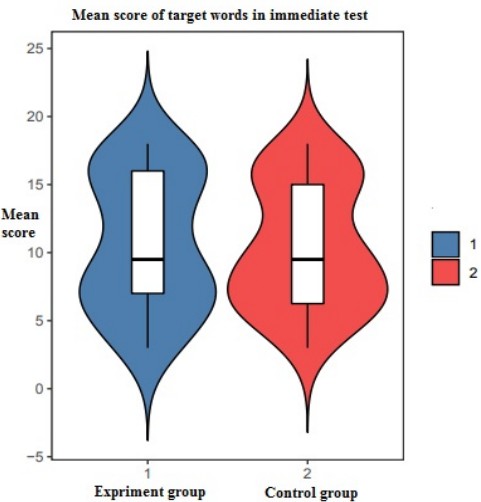

**Figure 2.** Target words in immediate test.

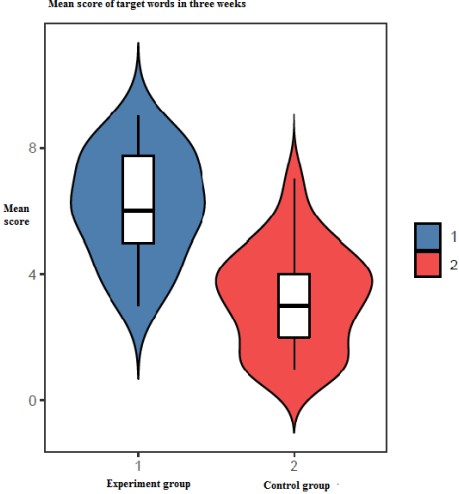

**Figure 3.** Target words in three week test.

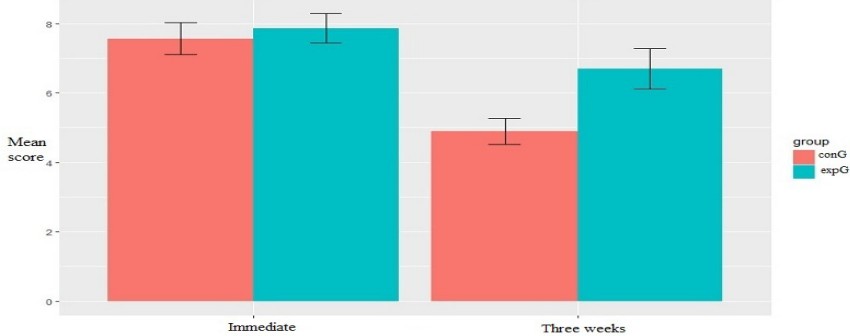

**Figure 4.** Comparison of vocabulary test scores between two tests.

It can be seen from Figure 4 that there was no obvious difference in immediate test, but the results in the experimental class were better than that of the control class after three-week experiment.

### 4.2. Results of the Contents Test

The descriptive statistics analysis of the manual's content tests in the experimental and control classes is also shown in Table 4. From Table 4, the results in the manual's content test implemented immediately after the teaching experiment and after three weeks were similar to those of the target test, close to those of the experimental class and the control class (7.53 vs. 7.33 $p = 0.55$, see Figure 5). After three weeks, the results of the experimental class and the control class were greatly reduced, but it seems that the decline of the control class is more significant (5.80 vs. 3.43 $p < 0.001$, see Figure 6).

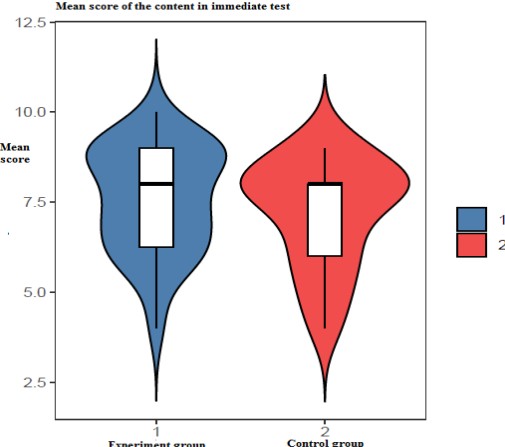

**Figure 5.** The content in the immediate test.

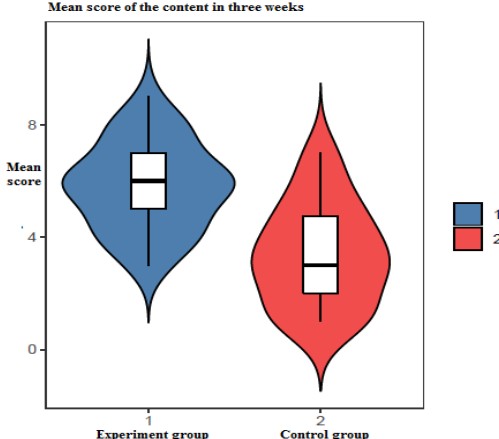

**Figure 6.** The content in the three week test.

Similarly, the immediately posttest and three week posttest were also conducted to test whether there were significant differences of content scores and whether the time had affected them differently, the ezANOVA ( ) function in R language was also used to analyze $2 \times 2$ mixed-design ANOVA. The group was an inter-group factor with two levels (experimental vs. control class), and the time was an intra-group factor with two levels too (immediate test vs. three weeks later). The results showed that the group had a main effect ($F_{(1, 78)} = 17.51$, $p < 0.001$, $\eta^2 = 0.121$), that is, the content test in the experimental class was better than that of the control class. The time also had a main effect ($F_{(1, 78)} = 80.43$, $p < 0.001$, $\eta^2 = 0.412$), that is, the immediate test scores were better than the delayed test scores. Furthermore, there was also a significant interaction effect between the groups and the time ($F_{(1, 78)} = 10.36$, $p < 0.001$, $\eta^2 = 0.071$), The effect of time on the control class was stronger than the experimental class. These results can also be seen in Figure 7.

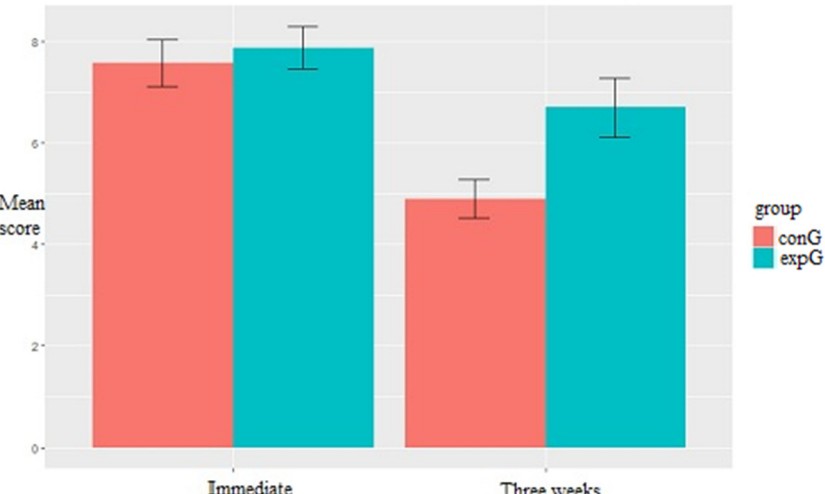

**Figure 7.** Comparison of the content test scores between two tests.

It can be seen in Figure 7 that there was no significant difference between the experimental and control classes for the content test immediately posttest (the error bars of the two groups highly overlapping). The results in the experimental class were better than the control class after three weeks (the error bars of the two groups without overlap).

### 4.3. Results of the Target Words and the Content

The purpose of this research is to investigate the impact of the "Interactive Learning Model" English teaching model on foreign language learners' vocabulary acquisition. Therefore, it is necessary to further examine the relationship between the content and the vocabulary performance. We used the lmer ( ) function of the lme4 package in R, with vocabulary as the dependent variable, group and time as predictors, and content as the covariate. We fitted the mixed-effect model by examining the random effect structure and fixed effect structure in the model separately. Fixed effects are, essentially, the predictor variables. It is the effect of group and time after accounting for random variability. Random effects are best defined as noise in the data (uncontrollable variability within the sample). Subject level variability is often a random effect.

The results show when the scores of the content variable is entered into the model, and the time and group have no interaction effects. By using the summary ( ) function to view the regression coefficient structure of the model, we find that the group variable still has no main effect (F (1, 79.91) = 0.01, $p$ = 0.94), but the time variable has the main effect (F (1, 98.57) = 258.36, $p$ < 0.001). The content variable shows a predictive effect (F (1, 155.62) = 26.95, $p$ < 0.001), when other variables remain the same, the score of the content in the case of our model here adds each unit, that is, a point, vocabulary score will increase significantly by 0.71 points (β = 0.71, SE = 0.14, $t$ = 5.19, $p$ < 0.001). This means that on the delayed posttests (the time variable had the main effect), the participants' performance in the content test had a very different effect on the scores in the vocabulary test of the two groups.

## 5. Discussion

### 5.1. Incidental Vocabulary Acquisitions

From the above experimental results, we can see that students in both the experimental class and the control class have acquired lexical knowledge obviously, proving that "reading is beneficial", and further confirming that foreign language reading is an effective way for incidental vocabulary acquisition. Incidental vocabulary learning, also known as implicit vocabulary learning, occurs when the mind is concentrated elsewhere, such as on comprehending a written text or understanding spoken material. One of the premises

of implicit vocabulary learning is that new words should not be presented in isolation and should not be learnt by mere rote memorization. It follows that new vocabulary items should be presented in contexts rich enough to provide clues to meaning, and that learners should be given multiple exposure to items they are supposed to learn [37]. Lack of exposure is a common problem facing language learners. A good way to combat this problem is to expose learners to extensive reading which offers broad exposure to the target language and is second only to acquiring the language by communicating with its native speakers [36]. Therefore, in the "Interactive Learning Model", learners were exposed to the target language, and at the same time, they acquired new words by exchanging information. Furthermore, input is meaningful and engaging because texts are chosen by readers in accordance with their preferences, and so provide a medium for attaining individual pleasure and enlightenment [38].

An analysis of the long-term memory effect (three weeks) on lexical acquisition shows that interactive reading significantly affects incidental vocabulary acquisition. In the immediate posttest, the scores of the experimental class and the control class were very close, but they differed significantly after three weeks. The time variable profoundly influences the two classes differently. Considering that lexical tests primarily focus on word semantics, this influence is reflected in the long-term memory for lexical semantics. Therefore, the "Interactive Learning Model" teaching method offers a greater advantage on offline consolidation of lexical acquisition than a traditional "reading and teaching" method [30,31,39].

*5.2. Effects of Offline Consolidation*

Offline consolidation is also called overnight consolidation. After the new words are learned, the new words are strengthened in sleepiness, interacting and merging with the existing words in the brain, and gradually integrating into the psychological lexicon.

Concerning lexical acquisition, academics have distinguished between lexical configuration and lexical engagement. The former refers to the rapid acquisition of explicit vocabulary knowledge, while the latter refers to a slower process of interaction and fusion between newly learned words and existing words in the mental lexicon [29,40]. According to the interpretation of the complementary learning systems (CLS) proposed by Davis and Gaskell (2009), initial exposure to a new word will only lead to sparse representations in the short-term hippocampal system, consolidated to enhance long-term memory representations in neocortical memory [41,42].

To understand the advantages of the "Interactive Learning Model" over the traditional "reading and teaching" model in lexical acquisition, especially offline consolidation of lexical semantics, it may still be necessary to discuss the core differences between the two modes. According to the experiments, the "common responsibility" or "common intension" driven by the "knowledge gap" promotes different "episodes" of two groups' foreign language competence, which makes all difference between the "Interactive Learning Model" and the traditional model. Although the traditional model and the "Interactive Learning Model" are initially driven by teaching tasks (such as telling students to perform both lexical and content tests, or submitting content summaries, etc. after the course is over), the driving force for students may not be the same. In the "Interactive Learning Model", the reading materials in each group are different from those of other groups, and the material in the test is dissimilar among six groups. Therefore, in order to complete the test smoothly, group members have to communicate with another five group members, carrying out meaningful exchange, otherwise it may be impossible to complete the test due to lack of relevant content. In order to carry out meaningful exchange, it is necessary to form a strong "social context" based on "common responsibility" and "common intension".

*5.3. Effects of Social Context*

"Social context", aiming at adequate communication, will further promote the emergence of two episodes in foreign language development, namely, LREs and IR [20,26]. In

this study, LREs and IR are complementary. LRE is mainly related to language, such as meaningful negotiation, speech repair, questioning, and correcting language use in discourse interaction. IR usually takes information request as the core purpose, relating to the content of reading. As students are required to use English in communication in the study, IR is more likely to promote unconscious language use. The emergence of "shared responsibility", LREs and IR were confirmed in our later interviews. For example, in answering how to find the meaning of the target pseudo-words in the reading materials, almost every interviewee in the control group replied "basically depending on guess", including "conjecture based on context", "guesses based on suffixes and affixes" or "association based on articles contents". Although the students' questioning and the teachers' explanations are central to the design of the teaching model in control class, few interviewees mentioned discussing with or asking questions about the words' meaning from teachers, and the teacher seemed to play the role of a lonely knowledge disseminator. However, almost every interviewee in the experimental class mentioned that they would "talk to each other" or "communicate with other group members" when they were uncertain about the meaning of the words. Moreover, they also showed a strong sense of responsibility, saying that each group took responsibility for a part of the reading material when communicating. For example, interviewees from the experiment class mentioned:

> "As for unfamiliar words introduced by other group members, we would ask directly. Sometimes after discussion with each other over the uncertain problem, our minds were filled with visions and ideas, and we were also impressed. Sometimes we had to explain the word for others and would feel humiliated if we could not get ourselves across, so we thought a bit more about the part we were responsible for."

These quotes show how their intra- and intergroup interactions helped them learn, as well as how they understood their responsibilities. These students' positive feelings about their peer interactions and their responsibilities might indirectly reflect the effectiveness of the interactions.

### 5.4. Effects of Interactive Reading

Interactive reading is an effective interactive learning model. At first, we read independently, and then read it together again according to our own understanding.

"Cooperation, communication, discussion, responsibility" and other features are imprinted everywhere in the "Interactive Learning Model". Studies have shown that the use of conscious and unconscious language resources, led by LREs and IR, are more likely to facilitate students to process foreign languages (reading, listening, and speaking) in a deeper way. For example, by refining and practicing, they can connect language with content, images or more in-depth analysis, which would enhance the learner's implicit learning ability [43].

If only from the names of the two teaching models, the "Interactive Learning Model" has a distinct step of discussion achieving better results compared with the traditional "reading and teaching" model. Some people think that the "Interactive Learning Model" is better than the traditional model because the discussion steps are added in the experiment. In fact, from the detailed introduction in the study, we see that the concept of "more" and "less" in the two models must be viewed dialectically. Simply regarding the better effect as the result of discussion in the "Interactive Learning Model" could be a too intuitive hypothesis. Seemingly, the "Interactive Learning Model" has more discussion steps than the traditional models, but it lacks several steps in this. For instance, the time given to reading in the "Interactive Learning Model" is significantly less than that in the traditional mode (15 min vs. 30 min). In addition, because ten target pseudo-words are evenly distributed in the whole reading material of the traditional model, students are more likely to be exposed to these words repeatedly while reading. Prior studies have revealed that with a longer reading time, the repeated exposure to the target word will significantly improve its acquisition [38]. From this point of view, the traditional model is more advantageous.

Therefore, the long-term memory advantage we have observed in the "Interactive Learning Model" cannot simply be attributed to the distinction in "more" or "less" questions. Rather, we should return to the element of adequate meaningful communication in the "social setting" promoted by the "Interactive Learning Model".

Students' meaningful communication has been also reflected in the above statistical results. The content test score in the statistical test, the group factor, has the main effect without considering any other factors, that is, the content test results in the experimental class is better than that of the control class. The group factor also displays a significant interaction with the time factor, that is, students in the experimental class report significantly better memory of the content than those in the control class in the long run. As mentioned above, for the experimental class to achieve better content test results, only sufficient meaningful communication within and between groups, especially the latter, can take effect. Otherwise, the "knowledge gap" resulting from the reading material will significantly lower their content test scores.

In other words, meaningful communication entails better content, longer memory and strengthens offline consolidation of the acquisition of the word. This can also be corroborated by previous studies on second language acquisition. In 2015, Applied Linguistics, the world's leading journal of applied linguistics, released a special issue that attempted to reflect on and redefine the connotation and the research scope of applied linguistics. Tarone (2015), in this issue, reviewed and summarized some generally accepted conclusions in the second language acquisition field in the past 90 years [44]. Among them, he mentioned two points: (1) the explicit knowledge, the L2 grammatical rules, acquired by learners has little to do with their language proficiency and their ability to communicate in L2. In other words, providing an L2 learner with explicit descriptions of L2 rules does not in itself make the acquisition of the ability to use these rules, in particular, the ability to use the second language as a medium for meaningful communication. (2) Successful second language acquisition is essentially "social" or "communicative" [38]. A second language acquisition process is fundamentally driven by the learners' involvement in social and communicative settings, as well as having meaningful communication with others. Our results further support Tarone's conclusion.

The above results also demonstrate that in both the experimental class and the control class, learning content has positive effects on vocabulary acquisition. According to the results of mixed-effect models, each 1 point increase in content test scores will result in a significant 0.71-point increase in vocabulary test scores. For the "Interactive Learning Model", the content test results indicate full meaning communication, while for the traditional model, the content test scores represent the depth of understanding of reading. It can be concluded that in-depth reading is more conducive to vocabulary acquisition. Foreign language teachers are right to set the goal of "reading for the sake of understanding". However, we did not observe the interaction effect between the content and the group in the mixed-effect model, that is, the effect of the content on vocabulary acquisition is not different between the two classes. It can be seen that the effect of long-term memory on vocabulary acquisition in the "Interactive Learning Model" depends on other factors, such as the enhancement of learners' implicit learning ability caused by the combination of language and content, image or deeper analysis through refinement, and practice as summarized above.

## 6. Pedagogical Implication

The purpose of this study was to investigate whether there were significant differences between the experimental class and the control class after the immediate posttest and after a three-week posttest, and whether the time had affected them. Via a mixed-effect model, vocabulary worked as a dependent variable, groups and time as predictive variables, content as a covariance variable. The experimental result show that the experimental group retains more vocabulary than its counterpart after three weeks. That is, students forget less when they participate in interactive discussions. An information-gap activity allows

students to pool information from their peers and simulates real-life situations. Thus, it enables students without any knowledge to obtain that information from those who do. Due to the fact that the students' comprehension of the five texts that they had not read in the "Interactive Learning Model" class depended on the oral summaries provided by the other students who had read the texts, their comprehension may have been influenced by the quality of the oral summaries. Therefore, it is vital that students receive training on how to prepare effective oral summaries in the "Interactive Learning Model" class, and oral summaries also improve the effectiveness of reading. Wajnryb's (1988) "read-ask-tell" and "jigsaw reading" methods assign each student or group a different text of short length, creating information gaps. Furthermore, the activities outlined above will assist learners in developing shared intentionality and shared responsibility, both of which are essential for meaningful negotiation in foreign language learning [25].

The experiment results also showed that in our model, when the content score adds a unit, or a point, the vocabulary score will rise significantly by 0.71 points, and the time variable has the main effect. It means that the "Interactive Learning Model" results in better content and lexical retention in the experimental class than in traditional class, especially three weeks after the class. Due to the correlation between retention of L2 word knowledge and reading material content knowledge, teachers and students explore interesting content while students are engaged in appropriate language-dependent activities. When students are exposed to a considerable amount of language while interactively learning, language learning is successful because the language is relevant to the learner's needs. Moreover, a link among vocabulary and content also provides support for content-based instruction (CBI) for EFL learning, rather than instruction that emphasizes mainly language skills and knowledge [45]. Moreover, CBI should engage college EFL students more than language knowledge-bound instruction when it comes to language learning.

## 7. Conclusions

According to the above statistical results, three conclusions can be made: (1) the advantage of the "Interactive Learning Model" in vocabulary acquisition is mainly reflected in the long-term memory of vocabulary, in particular, the long-term memory of lexical semantics; (2) the learning of the content (or reading content) will profoundly affect vocabulary acquisition, especially the long-term memory of lexical semantics; (3) and the "Interactive Learning Model" helps students improve learning efficiency.

The "Interactive Learning Model" seems to be common in terms of terminology, and it seems also a cliché when applied to teaching. In fact, this is not the case. As stated at the beginning, the "Interactive Learning Model" is refined based on the long-term theoretical and practical achievements. The language used to describe the foreign language teaching model is simple, clear, and direct, but it conveys the complex cognitive and psychological process involved. Its actual content is far more than the terminology itself literally, because it emphasizes the formation of "shared responsibility" or "shared intentionality", which helps to construct the "social setting", helping to promote the occurrence of the two "episodes" of language development, and further forming rich interactive resources and available language resources.

In essence, the "Interactive Learning Model" is one combining content and language. It takes into full account the characteristics of college English teaching in Chinese foreign language environment, aiming to provide the learners with a "social" or "communicating" setting, in which they are exposed to the language environment and learn to use FL to transmit meaning in oral interaction with others. The "Interactive Learning Model" contributes to sustained reading development in a second language. It is imperative that teachers refine and continue the interactive learning instruction, replacing the traditional teaching method, without limiting it to a semester or academic year in order to enhance the sustainability of reading.

**Funding:** This work was founded by People's Republic of China, Ministry of Education (grant no. 18YJA740063).

**Institutional Review Board Statement:** All subjects gave their informed consent for inclusion before they participated in the study. The study was conducted in accordance with the Declaration of Helsinki, and the protocol was approved by the Ethics Committee of Xi'an Jiao Tong University Health Science Center (Project identification No. 202102200023, 27/2/2021).

**Informed Consent Statement:** Informed consent was obtained from all subjects involved in the study.

**Data Availability Statement:** The data that support the findings of this study are available on request from the author, upon reasonable request.

**Conflicts of Interest:** The author declares no conflict of interest. The funders had no role in the de sign of the study; in the collection, analyses, or interpretation of data; in the writing of the manuscript, and in the decision to publish the results.

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
