# Peer review of "An “Interactive Learning Model” to Enhance EFL Students’ Lexical Knowledge and Reading Comprehension"

_sustainability, doi:10.3390/su15086471_

Round 1

Reviewer 1 Report

In general

·         The article seems to have a clear and focused research question on the impact of interactive learning on vocabulary acquisition in foreign language learners. The methodology used, including the statistical analysis, appears to be sound and appropriate for the research question. The results of the study are clearly presented and discussed, and the implications of the findings for language teaching and learning are addressed.

·         Overall, the author seems to have a strong grasp of the relevant theories and concepts in the field of foreign language teaching and learning, and has demonstrated a good understanding of how to conduct and report on research in this area.

·         The author's writing is generally clear and well-organized, although there are some areas where the language could be improved for greater clarity and precision.

·         A more consistent and clear citation formatting is needed. In some cases, sources are cited within the text using only numbers, while in others, the author and year are included. Please, standardize this formatting throughout the paper.

1.      Introduction

·         The EFL acronym should be defined first before being used. Correct this please.

·         In the sentence "Reading is one of the important language skills, empowering individuals and enhancing societal development", consider adding a citation or a justification to support this claim.

·         In the sentence "However, historically EFL reading instruction has not been particularly effective in accomplishing a series of challenging learning tasks because EFL students has been taught silently reading or reading aloud", the phrase "EFL students has been taught" should be changed to "EFL students have been taught" to match the subject-verb agreement.

·         In the sentence "Therefore, students often consider it boring to read in English and find it hard to comprehend the text [3]", consider adding a citation or a justification to support this claim.

·         In the sentence "This being the case, this study investigates empirically the effectiveness of Interactive Learning Model on the comprehension and vocabulary gains of Chinese college EFL students, in comparison to traditional non-interactive reading instruction", consider rephrasing the beginning of the sentence for clarity. It may be clearer to say something like "Therefore, the present study aims to investigate empirically the effectiveness of the Interactive Learning Model on the comprehension and vocabulary gains of Chinese college EFL students, in comparison to traditional non-interactive reading instruction".

·         In the sentence "With reference to the interaction theory of language learning [5, 6] as well as the soci-ocultural theory of learning [7, 8], in Interactive Learning Model, intra-group and inter-group interactions after reading are involved, both spoken and written, aiming to provide the learners with a social setting, in which they exposed to the language environment and learn to use EFL to send meaning in oral interaction with others", consider breaking this sentence up into two or three shorter sentences for clarity.

·         In the sentence "Essentially, the Interactive Learning Model is an attempt to build “shared responsibility” or “shared intentionality” by creating “knowledge gap” of content and language", consider adding a citation or a justification to support this claim.

2.      Review of Literature

·         A brief introduction should be given in order to link things together. Please, insert.

2.1.  The Interaction Approach

·         In the second paragraph, it would be clearer to use the phrase "noticing gaps in their L2 knowledge" instead of "notice gaps in their L2 knowledge".

2.2.  Sociocultural Interaction Theory

·         In the paragraph discussing the "zones of proximal development", it might be helpful to explain the concept a bit more fully for readers who are not familiar with it. Additionally, the sentence "A teacher's role is to help learners maximize their potential by providing them with learning experiences in their zones of proximal development" could be revised for clarity.

·         In the sentence "Learners must engage in meaningful communication to fill in gaps in their content and language knowledge through the use of learning activities or tasks", it might be helpful to provide a bit more explanation of what is meant by "learning activities or tasks".

·         The sentence "Between-group activities also provide learners with additional opportunities to practice and reinforce their language skills" could be revised to clarify what is meant by "between-group activities" and how they differ from within-group activities.

2.3.  Interactive Teaching Approaches

·         In the sentence "Jigsaw reading approach emphasis that each part of the text to be read in a class is assigned to a group of students", "emphasis" should be changed to "emphasizes" to ensure subject-verb agreement.

·         In the sentence “This ER-plus approach described by Boutorwick et al”, the word “This” seems to be wrong. Please, correct.

·         When discussing the similarities among the three approaches, it would be beneficial to explicitly state what these similarities are, rather than just mentioning that they all involve interactive discussions to facilitate reading comprehension. This will help the reader to better understand the connections between the different approaches and how they relate to the theories of interaction learning and sociocultural learning.

2.4.  Lexical learning

·         The section’s title should start with a capital letter. Please, correct.

·         Some of the sentences are quite long and complex, which may make it difficult for readers to follow the argument. It may be helpful to break up longer sentences into smaller, more digestible chunks.

3.      Methodology

·         A brief introduction should be given in order to link things together. Please, insert.

3.1.  Experimental design

·         The authors must start the passage with a clear statement of the research question or hypothesis that the study aimed to address. This will help readers understand the context and purpose of the study from the outset.

·         It would be helpful to clarify what is meant by "interactive reading approach" and "traditional teaching approach" in more detail, to help readers understand the specific teaching methods used in each group.

·         Consider breaking up the long sentence that describes the traditional teaching approach into two shorter sentences, for example.

3.2.  Participants

·         The caption of Table 1 should not be separated (in different pages) from the table.

·         The author must consider providing more information about the characteristics of the students in the study, such as their native language, as this could potentially affect their language learning abilities.

·         It might be helpful to provide more details about the teacher who taught the English writing classes, such as their qualifications and teaching experience.

3.3.  Materials

·         Please correct the number of the sub-section (3.3).

·         The explanation of why the Florida Travel Manual was chosen as the reading material could be improved by providing more information about the specific language features and vocabulary that were targeted in the manual.

·         In the second example there are a broken line. Please, remove.

·         The description of how the experimental and control groups were divided could be made clearer. Specifically, the author could explain why the experimental group was divided into six groups and what the purpose of having different materials for each group was.

·         It might be helpful to provide more information about the control group, such as how the pseudo-words were selected for the control group's reading material and what the purpose of the control group was.

3.4.  Experiment Process

·         A brief introduction should be given in order to link things together. Please, insert.

3.4.1.      The experimental process of experimental class

o    In step (3), consider adding "of the reading materials" after "summary" to make it clear what is being summarized.

o    In step (4), it might be clearer to say "After the group discussion, two members from each group were selected to rotate and communicate with members of other groups".

3.4.2.      The experimental process of the control class

o    In step 4 of the control class, it would be clearer to specify that students had a set amount of time to write their summaries, and how long that time was.

3.4.3.      Experiment material

o    In the sentence "Two sections were included in the vocabulary test, which focused on understanding the meaning of the words", it might be clearer to say, "Two sections were included in the vocabulary test, which aimed to test understanding of the meaning of words".

o    In the sentence "This section contained 20 items, 10 of which were target pseudo words and 10 of which were real English words selected randomly from the text", it would be clearer to specify whether the 10 target pseudo words were selected randomly from the text as well.

3.6.  Data collection

·         Please correct the number of the sub-section (3.6).

·         It would be helpful to clarify that the noise words are distractors that are not included in the scoring.

4.      Results

·         A brief introduction should be given in order to link things together. Please, insert.

4.1.  Results of target words test

·         Figure 1 already exist. Please correct the numeration of the figures and correct accordingly in the text.

4.2.  Results of the contents test

·         Figure 3 already exist. Please correct the numeration of the figures and correct accordingly in the text.

·         There are no Figure 5, but there are a Figure 6. Please correct the numeration of the figures and correct accordingly in the text.

4.3.  Results of the target words and the content

·         In the sentence "We fitted the mixed-effect model by examining the random effect structure and fixed effect structure in the model separately", consider specifying what exactly you mean by "random effect structure" and "fixed effect structure". This will help readers who may not be familiar with these terms to understand the methodology more clearly.

·         In the sentence "The content variable shows a predictive effect", consider clarifying what you mean by "predictive effect".

·         In the sentence "when other variables keep the same", consider rephrasing this to "holding all other variables constant" for clarity.

5.      Discussion

·         Some sentences and phrases could benefit from further clarification.

·         The use of apostrophe in "two groups’" is incorrect. Please, remove.

·         The author could further explain the concept of "implicit learning ability" and how it relates to vocabulary acquisition in the interactive learning model.

·         The author could discuss the potential limitations of the experiment, such as the specific characteristics of the study population or the duration of the experiment.

6.      Pedagogical implication

·         It would be helpful to explain briefly the lmer() function.

·         It would be helpful to define the "mixed-effect model", for readers who may not be familiar with these concepts.

·         It would be helpful to provide some specific examples of information-gap tasks and how they can be used to promote student-to-student interaction.

·         The statement "the experiment supports content-based instruction (CBI) for EFL students" could be expanded upon. Why does the experiment support this approach, and how can teachers incorporate CBI into their teaching?

7.      Conclusion

·         In the sentence "This is a type of foreign language teaching mode expressing the complex cognitive and psychological process in foreign language teaching and learning with simple, clear and direct words", the use of "foreign language teaching" twice in the same sentence could be simplified for clarity.

·         The phrase "It is imperative that teachers refine and continue the interactive learning instruction for a longer period of time" could be clarified. Does this mean that teachers should continue to use the interactive learning model for a longer period of time in each class session, or that they should use the model consistently over a longer period of time (e.g. throughout the semester or academic year)? Providing more specific guidance could be helpful.

References

·         The references used are old. Of the 40 references used only 6 are from the last 7 years. Please insert references that are more recent.

·         The reference list should have a common citation styles and it does not. Please, correct.

Author Response

Dear the reviewer, 

I would like to thank the reviewers for their very constructive comments. I’ve followed their comments/suggestions closely in revising our article. Below I describe in detail how I’ve responded to each of the reviewers’ comments. My response is in bold (black and red). My revision in the main document is in red.

In general

  • The article seems to have a clear and focused research question on the impact of interactive learning on vocabulary acquisition in foreign language learners. The methodology used, including the statistical analysis, appears to be sound and appropriate for the research question. The results of the study are clearly presented and discussed, and the implications of the findings for language teaching and learning are addressed.
  • Overall, the author seems to have a strong grasp of the relevant theories and concepts in the field of foreign language teaching and learning, and has demonstrated a good understanding of how to conduct and report on research in this area.
  • The author's writing is generally clear and well-organized, although there are some areas where the language could be improved for greater clarity and precision.
  • A more consistent and clear citation formatting is needed. In some cases, sources are cited within the text using only numbers, while in others, the author and year are included. Please, standardize this formatting throughout the paper.

Response: I thank the reviewer for the very positive comments. We also thank the reviewer for pointing out our limitations.

 Introduction: The EFL acronym should be defined first before being used. Correct this please.

Response: I thank the reviewer for the comment. I have changed the acronym “the EFL” accordingly, English as foreign language learning (EFL).

  •  

 In the sentence "Reading is one of the important language skills, empowering individuals and enhancing societal development", consider adding a citation or a justification to support this claim.

Response: I thank the reviewer for the comments. I added the citation to support the claim: Modern society relies heavily on reading to promote individual development (Gao, 2022).

In the sentence "However, historically EFL reading instruction has not been particularly effective in accomplishing a series of challenging learning tasks because EFL students has been taught silently reading or reading aloud", the phrase "EFL students has been taught" should be changed to "EFL students have been taught" to match the subject-verb agreement. 

Response: I thank the reviewer for the comments. I changed the sentence accordingly.

 In the sentence "Therefore, students often consider it boring to read in English and find it hard to comprehend the text [3]", consider adding a citation or a justification to support this claim.

I thank the reviewer for the comments. I added the citation to support the claim. “If there is lack of interactive learning among students, and teachers are rarely questions, students are not given the opportunity to think and participate, the classroom is very boring, and students are not interested (Chong & Tian, 2022).”

 In the sentence "This being the case, this study investigates empirically the effectiveness of Interactive Learning Model on the comprehension and vocabulary gains of Chinese college EFL students, in comparison to traditional non-interactive reading instruction", consider rephrasing the beginning of the sentence for clarity. It may be clearer to say something like "Therefore, the present study aims to investigate empirically the effectiveness of the Interactive Learning Model on the comprehension and vocabulary gains of Chinese college EFL students, in comparison to traditional non-interactive reading instruction".

Response: I thank the reviewer for the comments. I changed the sentence accordingly.

the sentence "With reference to the interaction theory of language learning [5, 6] as well as the soci-ocultural theory of learning [7, 8], in Interactive Learning Model, intra-group and inter-group interactions after reading are involved, both spoken and written, aiming to provide the learners with a social setting, in which they exposed to the language environment and learn to use EFL to send meaning in oral interaction with others", consider breaking this sentence up into two or three shorter sentences for clarity.

Response: Yes, the sentence is a bit long and complicated. I broke them into several sentences. In Interactive Learning Model, we refer to the interaction theory of language learning [5, 6] as well as the socio-cultural theory of learning [7, 8]. Students interact both intra-group and inter-group after reading, both orally and in writing, ensuring a social atmosphere. Through exposure to the language environment and use of EFL in oral interactions, they develop the ability to communicate and send meaning in oral interaction.

In the sentence "Essentially, the Interactive Learning Model is an attempt to build “shared responsibility” or “shared intentionality” by creating “knowledge gap” of content and language", consider adding a citation or a justification to support this claim.

Response: I thank the reviewer for the suggestions. I added the two citations to support the claim. “Interaction is one of the characteristics of socially shared knowledge (Abildsnes, Flottorp & Stensland (2021).” “Content and language learning are increased by bridging the knowledge gap (Morton,2018)”

  1. Review of Literature: A brief introduction should be given in order to link things together. Please, insert.

Response: I thank the reviewer for the suggestions. I added the brief introduction at the beginning of literature. “The interactive reading model integrates three approaches or theories: interaction theory, sociocultural interaction theory, and lexical learning theory. Each one will be reviewed in literature section.”

2.1 The Interaction Approach:  In the second paragraph, it would be clearer to use the phrase "noticing gaps in their L2 knowledge" instead of "notice gaps in their L2 knowledge".

Response: I thank the reviewer for the suggestions. I changed the sentence accordingly.

2.2 Sociocultural Interaction Theory: in the paragraph discussing the "zones of proximal development", it might be helpful to explain the concept a bit more fully for readers who are not familiar with it.

Response: I thank for the reviewer’s suggestions. I gave the additional explanation for ZDF, making readers understand easily. that is, the distance between the actual developmental level and the level of potential development” (Vygotsky, 1978). Vygotsky’s theories stress the fundamental role of social interaction in the development of cognition (Vygotsky, 1978), as he believed strongly that community plays a central role in the process of “making meaning.

Additionally, the sentence "A teacher's role is to help learners maximize their potential by providing them with learning experiences in their zones of proximal development" could be revised for clarity.

Response:I thank for the reviewer’s suggestion. I rewrote the sentence to make it clarity.  

In order to maximize learners' potential, a teacher needs to provide them with learning experiences.

The “Learners must engage in meaningful communication to fill in gaps in their content and language knowledge through the use of learning activities or tasks", it might be helpful to provide a bit more explanation of what is meant by "learning activities or tasks".

Response: I thank for the reviewer’s suggestion. I added one example to explain the learning activities task. “for example oral communication to get what they don’t know the content.”

The sentence "Between-group activities also provide learners with additional opportunities to practice and reinforce their language skills" could be revised to clarify what is meant by "between-group activities" and how they differ from within-group activities.

Response: I thank for the reviewer’s suggestion. I rewrote the sentence and use the words intra and inter to replace the words tween and within. The students participated in a group discussion in a given group after reading the material because they were lack of relevant content and linguistic knowledge. Furthermore, they will exchange information with other groupmates to gain additional information. Therefore, In order to successfully complete their communicative activities, both intra-group and inter-group activities are sometimes needed.

    2.3. Interactive Teaching Approaches:  In the sentence "Jigsaw reading approach emphasis that each part of the text to be read in a class is assigned to a group of students", "emphasis" should be changed to "emphasizes" to ensure subject-verb agreement.

Response: Thank the reviewer’s suggestion. I changed the word accordingly.

  •  

 In the sentence “This ER-plus approach described by Boutorwick et al”, the word “This” seems to be wrong. Please, correct.

Thank the reviewer’s suggestion. I made the mistakes in the sentence. “According to Boutorwick et al. (2019)[10], this ER-plus approach has the following steps.”

      When discussing the similarities among the three approaches, it would be beneficial to explicitly state what these similarities are, rather than just mentioning that they all involve interactive discussions to facilitate reading comprehension. This will help the reader to better understand the connections between the different approaches and how they relate to the theories of interaction learning and sociocultural learning.

Response: Thank the reviewer’s suggestion. I added the difference and similarity among three approaches. “Despite the differences in procedures and activities, all three approaches emphasize interactive discussions among students to facilitate reading comprehension. Interaction learning theory and sociocultural theory of learning are both driven by or reflect the importance of meaningful student interaction.”

2.4. Lexical learning: the section’s title should start with a capital letter. Please, correct.

Response: Thank the reviewer’s suggestion. I capitalized these letter at the beginning.

Some of the sentences are quite long and complex, which may make it difficult for readers to follow the argument. It may be helpful to break up longer sentences into smaller, more digestible chunks.

Response: Thank the reviewer’s suggestion. I rewrote the following sentences accordingly.

“Using a variety of research designs and learning achievement measures, reading has been shown to positively impact both quantity and quality of lexical learning, including breadth and depth.”

In various research designs and learning achievement measures, reading can have a positive impact on lexical learning. It has been shown to improve both the quantity and quality of lexical learning, as well as breadth and depth.

“The “Interactive Learning Model” for foreign language teaching is grounded on the theoretical and practical results from academia, with the focus on the language environment of China’s foreign language teaching and the colleges’ students who had received English education for many years in standard classroom teaching environment.”

"Interactive Learning Model" is based on theoretical and practical results from academia. This study focuses on the language environment of China's foreign language teaching and the colleges’ students who have received traditional classroom instruction for many years.

And the “shared responsibility” and “shared intentionality” caused by the “knowledge gap” on both the contents and the language are the catalyzes for the “meaningful” interaction based on oral communications [18].

The "shared responsibility" and "shared intentionality" are caused by the "knowledge gap" both on the contents and the language. Through "meaningful” interaction, they facilitate oral communication.

The essential part of the “Interactive Learning Model” can be shown in the following figure though the details of implementation may vary according to specific circumstances and class requirements.

The essential part of the “Interactive Learning Model” can be shown in the following figure Implementation details may vary depending on specific circumstances and class requirements.

As is shown in Figure 1, the “Interactive Learning Model” first and foremost emphasized the succession of the Interaction Hypothesis, reflecting oral discussion and communication in the central role.

As is shown in Figure 1, the “Interactive Learning Model” first and foremost emphasized the succession of the Interaction Hypothesis. The central role is played by oral discussion and communication.

the reading is set as an important part, even as the starting point of learning because reading prepares for followed the summary writing and inner-group/inter-group discussion, meanwhile the reading should have certain breadth and depth and be based on contents.

Reading is included as an important part, even as the first step to learning the subject, as it prepares participants for the summary writing and between-group/within-group discussions that follows. In the meantime, content-based readings should be based on a certain breadth and depth.

that is, apart from emphasizing social interaction, it also emphasizes the “meaningful” communication in the inner-group/inter-group (between group/within group) interaction to achieve the in-depth understanding and extension of the reading contents (surely the deep level processing of the foreign language input is involved).

Apart from emphasizing social interaction, it also emphasizes "meaningful" communication between/within groups to support reading comprehension in-depth (surely the deep level processing of the foreign language input is involved).

that is, not only the individual should be responsible for the “knowledge” within the scope of their rights and obligations, but also the group members should take the joint responsibility (joint responsibility) to ensure that the interaction can be carried out smoothly and achieve the effect.

That is to say, the individual should not only be responsible for their "knowledge" within their rights or obligation, but also the group members to ensure that interactions go smoothly.

in this case, they asked for information from the peers of the same group and the neighbors group, which ignites the knowledge “Epistemic Search Sequence” (ESS), and finally they work together to solve the emerging “knowledge gap” with the team members

 The group members asked the peers and neighbors for information, which ignited the

"Epistemic Search Sequence" (ESS). Finally, they worked together to solve the problem.

In model design, the Interactive Learning Model not only tries to trigger various interactive resources and language resources through LREs and IR to provide opportunities for learner language development, but also fully considers the characteristics of college English teaching in Chinese and foreign language environments.

Learning resources and language resources can be triggered via LREs and IR to provide opportunities for learners to develop their language skills. It also takes into account the unique characteristics of college English teaching in Chinese and foreign language environments.

  1. Methodology: A brief introduction should be given in order to link things together. Please,

insert.

    Response: Thank the reviewer’s suggestion. I added the brief introduction.

To examine the effectiveness of “Interactive Learning Model” via the experiment class and the control class, the study invited 40 non-English major graduate students in the experiment and control class. In this class experiment, one class was taught with “Interactive Learning Model” and the control class was instructed with a traditional approach involving no student interaction. After the experiment, students in two classes took two posttests: immediate posttest and three weeks later posttest and were interviewed after immediate posttest.

3.1. Experimental design: The authors must start the passage with a clear statement of the research question or hypothesis that the study aimed to address. This will help readers understand the context and purpose of the study from the outset.

Response: Thank the reviewers’ suggestions. I added two research questions to state my research purpose.

  1. What is the impact of the “interactive reading approach” on EFL learners' content learning as measured by immediate and delayed posttests?
  2. What is the impact of the “interactive reading approach” on EFL learners' vocabulary learning as measured by immediate and delayed posttests?

  It would be helpful to clarify what is meant by "interactive reading approach" and "traditional teaching approach" in more detail, to help readers understand the specific teaching methods used in each group. Consider breaking up the long sentence that describes the traditional teaching approach into two shorter sentences, for example.

Response: Thank for the reviewer’s suggestion. “The interactive reading approach drew on practices from the various interactive reading approaches, such as "jigsaw reading," "read-ask-tell," and "ER-plus." Besides group discussion and writing activities, the interactive reading approach were driven by filling information/knowledge gaps similar to those found in the “jigsaw reading” and “read-ask-tell” approaches.” ....Detailed information about the two approaches and their procedures can be found under the subsection "Procedures".

3.2 Participants: the caption of Table 1 should not be separated (in different pages) from the table.

Response: Yes, thank for the reviewer’s suggestion. I combined them together.

The author must consider providing more information about the characteristics of the students in the study, such as their native language, as this could potentially affect their language learning abilities.

Response: I give more detailed information about the characteristics of the students. As Chinese students, they speak Chinese as their native language.

It might be helpful to provide more details about the teacher who taught the English writing classes, such as their qualifications and teaching experience.

Response: Thank for the reviewer’s the teacher with a doctorate had extensive experience teaching writing.

3.3. Materials: Please correct the number of the sub-section (3.3).

Response: Thank for the reviewer’s suggestion, I corrected the mistakes.

      The explanation of why the Florida Travel Manual was chosen as the reading material could be improved by providing more information about the specific language features and vocabulary that were targeted in the manual.

Response: I selected the Florida Travel Manual based on the following reasons. The manual is an authentic English text. The vocabulary level is appropriate, because 85.6% words in manual are from the General Service Word List (GSL), including the 2000 most commonly used English words. Participates (students) passed the College English Test (Band 4) administered by the Ministry of Education, a required English test for college students. In accordance with the Ministry of Education's information about CET-4, passing this exam requires a vocabulary size of 4,200. It means that the manual is appropriate for the participants and the participants have enough vocabulary sizes.

In the second example there are a broken line. Please, remove.

Response: Thank your reviewers’ suggestions. I removed the broken line.

The description of how the experimental and control groups were divided could be made clearer. Specifically, the author could explain why the experimental group was divided into six groups and what the purpose of having different materials for each group was.

  • Response: Thank for the reviewer’s suggestion. I added the reason why the group was divided into 6 parts. “In experimental class, the 18 pages of material were divided into 6 parts from the beginning to the end. A 6-part reading text (each with three pages plus the cover page of the manual) was then distributed evenly among the six groups. Therefore,40 students were also divided into 6 groups . In the first four groups, there are seven students each, and in the last two, there are six students each. In other words, each group had only one-sixth of the text's information.”

 It might be helpful to provide more information about the control group, such as how the pseudo-words were selected for the control group's reading material and what the purpose of the control group was.

Response: Thank for the reviewer’s suggestion. “In the control class, students were not divided into any groups. They all received the entire 18-page reading text, unlike students in experiment class, who were given parts of the reading text. The 18-page reading text in the control class was the same as in the experiment class, with pseudo-words replacement.”

3.4. Experiment Process: A brief introduction should be given in order to link things together. Please, insert.

Response: I added the brief introduction in experiment process. Experiment and control classes both involve four steps, but they concentrate on different aspects. The experimental class emphasizes interactive teaching methods, for instance, intra-group and inter-group discussions; the control class, however, is taught in a traditional manner, for example by means of teaching-learning without any interaction.  

3.4.1. The experimental process of experimental class

o    In step (3), consider adding "of the reading materials" after "summary" to make it clear what is being summarized.

o    In step (4), it might be clearer to say "After the group discussion, two members from each group were selected to rotate and communicate with members of other groups".

Response: Thank for the reviewers. I changed accordingly.

3.4.2. The experimental process of the control class

o    In step 4 of the control class, it would be clearer to specify that students had a set amount of time to write their summaries, and how long that time was.

Response: The summary writing was asked to finish in 20 min.  

“Writing. Students were required to write a summary in 20 min., describing the content of the material members have read, and then handing in them. The whole process is also about 65 minutes.”

3.4.3.      Experiment material

o    In the sentence "Two sections were included in the vocabulary test, which focused on understanding the meaning of the words", it might be clearer to say, "Two sections were included in the vocabulary test, which aimed to test understanding of the meaning of words".

Response: Thank for the reviewers. I changed accordingly.

o    In the sentence "This section contained 20 items, 10 of which were target pseudo words and 10 of which were real English words selected randomly from the text", it would be clearer to specify whether the 10 target pseudo words were selected randomly from the text as well.

   Response: Thank the reviewer’s question. I mean that the 10 target pseudo words mentioned before were not selected randomly. The real words were selected randomly from the text.

3.6.  Data collection

  • Please correct the number of the sub-section (3.6).

        Response: Thank the reviewer’s suggestion, and I corrected the mistakes.

  • It would be helpful to clarify that the noise words are distractors that are not included in the scoring.

Response: Thank the reviewers’ suggestion. I added the sentence in the text “these true words as distractors (fillers) were not included in the total scoring”.

  1. Result:  A brief introduction should be given in order to link things together. Please, insert.

Reponses: I added the brief introduction at the beginning of the Results. Statistical analyses show that interactive learning classes and traditional classes perform similarly on content and vocabulary immediately posttest; but in the delayed posttests, the Interactive learning class obviously outperformed the traditional class, that is the students in the experiment class forget less in vocabulary and content learning through intra/inter group discussion teaching model.

4.1.  Results of target words test:  Figure 1 already exist. Please correct the numeration of the figures and correct accordingly in the text.

Response: Thank the reviewer’s suggestion. The number of these figures has been reordered.

4.2.  Results of the contents test

  • Figure 3 already exist. Please correct the numeration of the figures and correct accordingly in the text.
  • There are no Figure 5, but there are a Figure 6. Please correct the numeration of the figures and correct accordingly in the text.

4.3.  Results of the target words and the content

  • In the sentence "We fitted the mixed-effect model by examining the random effect structure and fixed effect structure in the model separately", consider specifying what exactly you mean by "random effect structure" and "fixed effect structure". This will help readers who may not be familiar with these terms to understand the methodology more clearly.

Response: Thank the reviewer’s suggestion. I added the following sentence to help the clarification of these concepts. “Fixed effects are, essentially, the predictor variables. It is the effect of group and time after accounting for random variability. Random effects are best defined as noise in the data (uncontrollable variability within the sample). Subject level variability is often a random effect.”

 In the sentence "The content variable shows a predictive effect", consider clarifying what you mean by "predictive effect".

Response: Thank the reviewer’s question. In the mix-effect, the content as a covariance enter the model (see the following the mix-effect model). 

[the best model to fit is as follows: model <- lmer (vocabulary ~ group + time + content + (1|student), data=my data)]

I added the following sentence in text. “That is mean that on the delayed posttests (the time variable has the main effect), the participants’ performance in the content test had a very different effect on their scores on the vocabulary test of the two groups.”

 In the sentence "when other variables keep the same", consider rephrasing this to "holding all other variables constant" for clarity.

Response: Thank the reviewer’s suggestion. Yes, I mean other variables constant.

  1. Discussion
  • Some sentences and phrases could benefit from further clarification.

Response: Thank the reviewer’s suggestion. I give more explanation to make the readers’ clarification. See the following sentences.

The use of apostrophe in "two groups’" is incorrect. Please, remove.

Response: Yes, I removed the inappropriate expression.

  •  

The author could further explain the concept of "implicit learning ability" and how it relates to vocabulary acquisition in the interactive learning model.

Response: Thank the reviewer’s suggestion. I explained the concept in the following sentences: “Implicit vocabulary learning, also known as incidental vocabulary learning, occurs when the mind is concentrated elsewhere, such as on comprehending a written text or understanding spoken material. One of the premises of implicit vocabulary learning is that new words should not be presented in isolation and should not be learnt by mere rote memorization. It follows that new vocabulary items should be presented in contexts rich enough to provide clues to meaning and that learners should be given multiple exposure to items they are supposed to learn [35]. Lack of exposure is a common problem facing language learners; a good way to combat this problem is to expose learners to extensive reading which offers broad exposure to the target language and is second only to acquiring the language by living among its native speakers [36]. Therefore, in interactive learning model, learners was exposed to the target language, at the same time, they acquired new words by exchange information. Furthermore, input is meaningful and engaging because texts are chosen by readers in accordance with their preferences and so provide a medium for attaining individual pleasure and enlightenment [37].”

  • The author could discuss the potential limitations of the experiment, such as the specific characteristics of the study population or the duration of the experiment.

Response: Thank the reviewer‘s suggestion. I added some limitation in the text. “Although our results of our study compensate previous research findings, interactive reading instruction has not received enough empirical research, it would be important to carry out further studies that include both immediate and delayed posttests, especially studies with even larger sample sizes, to test the effectiveness of such reading instructional approaches on both content and vocabulary learning and retention. Or the experiment time is longer than 65 minutes, considering the activities the students are involved in during class is quite hectic, due to moving around and mingling with other groups. It is obviously that 65 minutes is considered very short.”

  1. Pedagogical implication: It would be helpful to explain briefly the lmer() function. It would be helpful to define the "mixed-effect model", for readers who may not be familiar with these concepts.

Response: Yes, mix-effect model is a bit complicated, easy to make readers confused. I added the explanation. “The purpose of this study was to investigate whether there were significant differences between the experimental class and the control class after the immediate test and after a three-week test, and whether the time had affected them. Via mixed-effect model, vocabulary worked as dependent variable, groups and time as predictive variable, content as covariance variable. The experimental result showed that the retention of vocabulary in the experimental group shows decrease of slowly compare to its counterpart after three weeks. That is, students forget less when they participate in interactive discussions.”

  • It would be helpful to provide some specific examples of information-gap tasks and how they can be used to promote student-to-student interaction.

Response: Thank the reviewer’s suggestion. I added “read-ask-tell” as the example to explain the information-gap. For example, Wajnryb's (1988) “read-ask-tell” and jigsaw reading method assign each student or group a different text of short length, creating information gaps. Furthermore, the activities outlined above will assist learners in developing shared intentionality and shared responsibility, both of which are essential for meaning negotiation in foreign language learning.

The statement "the experiment supports content-based instruction (CBI) for EFL students" could be expanded upon. Why does the experiment support this approach, and how can teachers incorporate CBI into their teaching?

Response: Thank the reviewer’s suggestion. The reason was added in the text. “The experiment results showed that the score of the content in the case of our model here adds each unit, that is, a point, vocabulary score will increase significantly by 0.71 points, and time variable have main effects. It means that the interactive learning model results in better content and lexical retention in experimental class than in traditional class, especially three weeks. Due to the correlation between retention of L2 word knowledge and reading material content knowledge, teachers and students explore interesting content while students are engaged in appropriate language-dependent activities. When students are exposed to a considerable amount of language while interactive learning, language learning is successful because the language is relevant to the learner's needs. Moreover, a link among vocabulary and content also provides support for content-based instruction (CBI) for EFL learning, rather than instruction that emphasizes mainly language skills and knowledge. Moreover, CBI should engage college EFL students more than language knowledge-bound instruction when it comes to language learning.”

  1. Conclusion: In the sentence “This is a type of foreign language teaching mode expressing the complex cognitive and psychological process in foreign language teaching and learning with simple, clear and direct words", the use of "foreign language teaching” twice in the same sentence could be simplified for clarity.

Response: Thank the reviewer’s suggestion. I rewrote the sentence again. “The language used to describe foreign language teaching model is simple, clear, and direct, but it conveys the complex cognitive and psychological process involved.”

  • The phrase "It is imperative that teachers refine and continue the interactive learning instruction for a longer period of time" could be clarified. Does this mean that teachers should continue to use the interactive learning model for a longer period of time in each class session, or that they should use the model consistently over a longer period of time (e.g. throughout the semester or academic year)? Providing more specific guidance could be helpful.

Responseï¼› “Here, it means that the instruction will not be limited to a semester or academic year. In order to make instruction more effective, an interactive teaching method should be used instead of a traditional teaching method. “To enhance the sustainability of reading, teachers must refine and continue interactive learning instruction to replace the traditional teaching method, without limiting it to a semester or academic year.”

References: The references used are old. Of the 40 references used only 6 are from the last 7 years. Please insert references that are more recent.

Response: Thank the reviewer’s suggestion. I replaced 13 references in 5 years, for example, reference 1, 2, 6…; I added 10 new references in 5 years (For example, Reference 3,5…).

  • The reference list should have a common citation styles and it does not. Please, correct.

Response: Thank the reviewer’s suggestion. These references were rearranged using the software.

Please see the attachment (My revision in the main document)

Best regards 

Lei Yang

Reviewer 2 Report

Please refer to the comments inside the manuscript.

Author Response

Dear the reviewer,

 I would like to thank the reviewers for their very constructive comments. I’ve followed their comments/suggestions closely in revising our article. Below I describe in detail how I’ve responded to each of the reviewers’ comments. My response is in bold (black and red). My revision in the main document is in red.

Abstract: Add the problem statement and significance

Response: I thank the reviewer’s suggestion. I added the research question and significance in the abstract.  “Research problems: To enhance the sustainability of reading, the article proposed the new teaching model-interactive learning model. What is the impact of the “interactive learning approach” on EFL learners' content and vocabulary learning?”

Significance: The significance of the research demonstrates “Interactive Learning Model” improves students' language learning motivation and offers the benefit of processing the foreign language more deeply and internalizing their knowledge through implicit learning.

Review of literature: Related with reading comprehension

Response: Thank the reviewer’s suggestion. Literature section includes three theories related to reading comprehension or interactive learning. For example, interaction hypotheses refer to the reading interaction hypotheses. To avoid confusion, I changed the 2.1 subtitle to “Interaction Hypothesis Theory and Comprehension Input”; 2.3 subtitle “Interactive Reading Instruction Programs”; 2.4 subtitle “Lexical learning and Reading”.

Methodology:Support with relevant reference

Response: Thank the reviewer’s suggestion. According to other researchers, I used pseudo-words for the test, and I added two references as well in text (e.g., Godfroid et al. 2013; Leach and Samuel 2007). In addition, I added the reference of Florida Travel manual (William 2017:46-78).

Discussion: categories the sub-themes

Response: Thank the reviewer’s suggestion. I added some subtitle for discussion.

5.1 Incidental vocabulary acquisitions

5.2 Effects of offline consolidation

5.3 Effects of social context

5.4 Effects of interactive reading

The references were updated

Response: Thank the reviewer’s suggestion. I replaced 14 references in 5 years, (for example, reference 1, 2, 6…); I added 10 new references in 5 years (for example 3, 5…).

Please see the attachment (My revision in the main document)

Best regards

Lei Yang

Reviewer 3 Report

Please refer to the document attached.
A few comments were highlighted here.
1. Abstract should be concise yet reflect the 'body' of the article. Hence it should have all the essential information of the paper.
2. The research design does not explain clearly, especially the way the author has done the post-test and post-test delayed. How did the author control both groups in terms of sample selection? This should be clarified.
How to control the homogeneity of the sample without a pre-test? What type of sampling design that used by the author?
3. How the author cites the reference in the text can also be improved.
4. Line 315-318 - How about the reliability and validity of this instrument? Do these instruments have undergone a pilot test? Or are the questions adopted from other research? If yes, was there any alteration before it was used in this research?
5. Line 329 - Why after three weeks? Some explanation and support would help here.

6. Line 382-384 - Can the author discuss why this happened? It could be because the class time is only 65 minutes, and it could be a difference if the class time is longer than 65 minutes considering the activities the students do during the class is quite hectic, i.e. moving around and mingling with other groups to get socialised, 65 minutes is considered a very short period.

7. Line 390-391 - What is the purpose of reducing students' performance in this study? Maybe some elaboration on how students may retain their competency should be highlighted here.

8. Line 569 - Better retention? This needs to be clarified. Because both groups decrease their output, maybe the way it explained should be "the experimental group shows slow decrement as compared to its counterpart, i.e., the control group"...or something like that. This is a humble suggestion.

More comments are in the attached document.

Author Response

Dear the reviewer, 

I would like to thank the reviewers for their very constructive comments. I’ve followed their comments/suggestions closely in revising our article. Below I describe in detail how I’ve responded to each of the reviewers’ comments. My response is in bold (black and red). My revision in the main document is in red.

  1. Abstract should be concise yet reflect the 'body' of the article. Hence it should have all the essential information of the paper.

Response: I thank the reviewer’s suggestions. I added research question and the significance. I revised the abstract.

  1. The research design does not explain clearly, especially the way the author has done the post-test and post-test delayed.

Abstract: how the design of the research? e.g., pre-test, post-test, post delayed test?

Response: I thank the reviewer’s question. Here I mean “two posttests are immediate posttest and three weeks posttest”. I added them in the abstract.

how many students involved? at what level? secondary? primary? etc.

Response: I thank the reviewer’s question. Here each class has 40 students, who are non-English major graduate students. In experiment method section, I described the subject.

How this result demonstrated? is there any instrument used?

Response: I thank the reviewer’s question. The results of statistical analyses indicate that through intra/inter group discussion method, interactive learning class and the control class makes similar improvements on both the content and vocabulary tests in immediately posttest; but in the delayed posttests, the Interactive learning class obviously outperformed the control class. In this study, I compared two different teaching models. I didn’t use other instrument.

Introduction: Say at first time need to be full name

Response: Thank for the reviewer’s suggestion. I changed into full name “English as foreign language learning (EFL)”

what is L2 knowledge?

Response: Thank the reviewer’s question. Here, L2 refers to the second language

ZPD?

Response: Thank the reviewer’s question. I give the explanation in section of “Sociocultural Interaction Theory”. The zone of proximal development (ZPD) refers to the distance between the actual developmental level as determined by independent problem solving and the level of potential development as determined through problem-solving under adult guidance, or in collaboration with more capable peers” (Vygotsky, 1978, p. 86)

jigsaw reading, red-ask-tell and ER Plus approaches?

Response: These are three reading teaching methods. I explained the three teaching methods in section “Interactive Teaching Approaches”.

what is the meaning of this? Before class (after class), since you have another session of After Class.

Response: Thank the reviewer’s question. In the interactive learning model, the term “before class” refers to “reading activities was conducted before class, because sometimes there is not enough time in class, the reading tasks can be scheduled before class. The term "after class" refers to the writing class scheduled after class.

IR, information retrieval

Response: Thank for the reviewer’s suggestion. I changed the word order.

  1. How to control the homogeneity of the sample without a pre-test? What type of sampling design that used by the author?

Response:Thank for the reviewer’s question. There is no difference in the level of subjects in the experiment class and the control class. The students in both classes (experiment class and control class) took the placement test before they began graduate studying. They were A level in the placement test. (90 scores and above in the placement were in A level; 80 score-90 score were in the B level, the rest of students were C level). I added the sentences to clarify the situation.

  1. How the author cites the reference in the text can also be improved.

Response: Thank the reviewer’s suggestion. I replaced 13 references in 5 years, for example, reference 1, 2, 6…; I added 10 new references in 5 years (For example, Reference 3,5…).

better put this in a table or flow chart as to show the difference between control and experimental group.

Response: Thank the reviewer’s constructive suggestion. Yes, it is good idea to design a table to show the difference.

Teaching classes

Teaching models/Time

Materials

Numbers

Experiment procedure

The experimental class

(A level)

Interactive Learning model (65 min.)

FTM (2018)

40 students)

(6 groups)

Reading      Intra-group discussion

Summary     Inter-group discussion

The control class

(A Level)

Traditional teaching model (65 min.)

FTM (2018)

40 students

Reading        Asking questions

Explaining questions       Writing

Notes: Florida Travel Manual (2018 edition)

Elementary school at what age? and now how many years they have learnt English?

Response: Thank the reviewer’s question. They all started learning English from the third grade (age 7-8) of elementary school. Years of English learning is about 13 years. I added the age 7-8 in the text.

True?

Response: Thank the reviewer’s question. Here, True refers to “6 True/False questions and 5 multiple-choice questions.” I added the information according.

  1. Line 315-318 - How about the reliability and validity of this instrument? Do these instruments have undergone a pilot test? Or are the questions adopted from other research? If yes, was there any alteration before it was used in this research?

Response: I thank the reviewer’s question. I did Cronbach’s alpha test for the two tests (the vocabulary test and the contest test). The reliability of this vocabulary test and the content test was 0.89 and 0.83 respectively. I added them in the text. .

  1. Line 329 - Why after three weeks? Some explanation and support would help (here.Why must in three weeks time? why not in 1 month or 2 weeks, etc.)

Response: Thank the reviewer’s question. I follow the practice of other researcher (Johanna & John, 1992).

  1. Can the author discuss why this thing happen? It could be because the class time is only 65 minutes, and it could be a difference if the class time is longer than 65 minutes considering the activities the students do during the class is quite hectic, i.e. moving around and mingling with other groups to get socialised, 65 minutes is considered a very short period.

Response: Thank the reviewer’s response. I discussed the results in discussion section. Two classes were conducted in 65 minutes, but three weeks' time interval was most likely the main reason for the difference of the result. The retention of vocabulary in experiment class is better after three weeks than in control class.

  1. What is the purpose of reducing students' performance in this study? Maybe a little bit of elaboration on how students may retain their competency should be highlighted here.

Response: Thank the reviewer’s question. Simply say, interactive learning model make students forget less in vocabulary and content because of offline consolidation, and shared responsible and so on. The purpose of reducing students' performance is to show that “three weeks” reduce students’ performance, that is, in mix-effect model, time produces the main effect, but interactive learning model retain their competency. I gave more explanations for these results in discussion and some suggestions in Pedagogical implication.

  1. Better retention? this is quit confusing. because both groups decrease their output. Maybe the way it explain should be "the experimental group shows decrease of slowly compare to its counter part, i.e., the control group"...or something like that. This is just a humble suggestion.

Response to reviewer: Thank the reviewer’s constructive suggestion. I didn’t express exactly. I changed accordingly.

Please see the attachment (the main document)

Best regards

Lei Yang

Reviewer 4 Report

In general, the paper has a good structure and style of presentation and is quite interesting.

As wishes: add some sources of literature "fresh" in recent years. It is advisable to add sources no later than 5 years ago.

is it possible to add several more sources (recent years) to the literature review in The Interaction Approach?

it is desirable to interpret all entered notations, hypotheses, hypothesis testing methods and results (for example, line 255: t 225 = -0.32, p = 0.75, line 372 F, etc., and in the following similar lines)

Author Response

Dear the reviewer,

I would like to thank the reviewers for their very constructive comments. I’ve followed their comments/suggestions closely in revising our article. Below I describe in detail how I’ve responded to each of the reviewers’ comments. My response is in bold (black and red). My revision in the main document is in red.

As wishes: add some sources of literature "fresh" in recent years. It is advisable to add sources no later than 5 years ago.

Response: Thank the reviewer’s suggestion. I replaced 14 references in 5 years, (for example, reference 1, 2, 6…); I added 10 new references in 5 years (For example, reference 3, 5…)  

is it possible to add several more sources (recent years) to the literature review in The Interaction Approach? input (or comprehensible input) is helpful to acquire second language

Response: Thank the reviewer’s suggestion. I added two sources in the interaction approach to help readers make sense of “input is helpful to acquire second language”.

“Interesting comprehension reading is very effective methods to improve reading, such as hearing interesting stories, because the pleasure in reading is more efficient than “study,” that is, more language are acquired per unit time.”[21]

“For example, the performance on standardized English tests was more closely tied to comprehension reading with negotiation than skill building (referring to conscious learning, output practice, and correction)” [22]

it is desirable to interpret all entered notations, hypotheses, hypothesis testing methods and results (for example, line 255: t 225 = -0.32, p = 0.75, line 372 F, etc., and in the following similar lines)

Response: Thank the reviewer’s suggestion. I deleted the sentence in the result section “P-values were reported in Table 4, rejecting the null hypothesis α = 0.05.” In my literature review section, I posed two research questions instead of hypotheses. In the result section, based the above two research questions, I provided relevant responses. Therefore, the sentence is appropriate here.

“The results were analyzed using R language. Table 4 is the descriptive statistics of the target words test (vocabulary test) and the content test results in immediate posttest and three-week posttest of the experimental class and the control class.”

Please see the attachment (the main document)

Best regards

Lei Yang

Round 2

Reviewer 3 Report

Dear author

In my opinion, you have done an excellent job in addressing the universal issue where English is a second language for non-native speakers. The model can be used in different countries maybe, and comparison should be assessed.